# Combined Effect of tDCS and Motor or Cognitive Activity in Patients with Alzheimer’s Disease: A Proof-of-Concept Pilot Study

**DOI:** 10.3390/brainsci14111099

**Published:** 2024-10-30

**Authors:** Cristina Fonte, Giorgia Rotundo, Valentina Varalta, Angelica Filosa, Ettore Muti, Cosimo Barletta, Elisa Evangelista, Massimo Venturelli, Alessandro Picelli, Nicola Smania

**Affiliations:** 1Neuromotor and Cognitive Rehabilitation Research Center, Section of Physical and Rehabilitation Medicine, Department of Neurosciences, Biomedicine and Movement Sciences, University of Verona, P.le L.A. Scuro 10, 37134 Verona, Italy; cristina.fonte@univr.it (C.F.); giorgia.rotundo@univr.it (G.R.); elisa.evangelista@univr.it (E.E.); alessandro.picelli@univr.it (A.P.); nicola.smania@univr.it (N.S.); 2Neurorehabilitation Unit, University Hospital of Verona, 37134 Verona, Italy; 3Section of Clinical Psychology, Department of Neurosciences, Biomedicine and Movement Sciences, University of Verona, 37134 Verona, Italy; angelica.filosa@univr.it; 4Monsignor Arrigo Mazzali Foundation, 46100 Mantua, Italy; ettore.muti@fondazionemazzali.it (E.M.); barletta.cosimo@libero.it (C.B.); 5Section of Movement Sciences, Department of Neurosciences, Biomedicine and Movement Sciences, University of Verona, 37134 Verona, Italy; massimo.venturelli@univr.it; 6Department of Internal Medicine, University of Utah, Salt Lake City, UT 84112, USA

**Keywords:** transcranial direct current stimulation, motor activity, cognitive stimulation, Alzheimer’s disease

## Abstract

(1) Background: Alzheimer’s disease (AD) accounts for 70% of dementia cases and with no effective pharmacological treatments, new rehabilitation methods are needed. Motor and cognitive activities and transcranial direct current stimulation (tDCS) have shown promise in stabilizing and enhancing cognitive functions. Objective: we want to investigate the effects of tDCS combined with motor or cognitive activity on cognitive functions in AD patients. (2) Methods: Patients with mild or moderate AD were randomized between anodic tDCS groups (MotA or CogA) and sham tDCS groups (MotS or CogS). They received two weeks of treatment (45 min, five days/week), with the first 15 min using tDCS stimulation on the dorsolateral prefrontal cortex. Cognitive assessments were conducted pre-treatment (T0), post-treatment (T1), and one week after (T2). (3) Results: Twenty-three patients were included. Statistical analysis showed significant differences between anodic tDCS groups (MotA + CogA) and sham tDCS groups (MotS + CogS) with advantages for the first in improving global cognitive status (*p* = 0.042), selective attention (*p* = 0.012), and sustained attention (*p* = 0.012). Further analysis indicated no differences between the two anodic tDCS groups between T0 and T1. (4) Conclusions: combined anodal tDCS with motor or cognitive activity could improve global cognitive state and attention, slowing cognitive decline in AD patients. The trial was registered on Clinical Trials: NCT06619795.

## 1. Introduction

As our global population gets older, the World Health Organization (WHO) predicts that the number of people living with dementia will increase significantly. In 2019, there were 55 million people with dementia, but by 2050, this number is projected to reach 139 million [1]. Alzheimer’s disease (AD) is known to be the most common form of dementia, contributing to 60–70% of cases (WHO, 2023), and it affects regions of the cerebral cortex, firstly the temporal and frontal lobes, and then progresses to the neocortex. Cognitive disabilities start with the clinical hallmark of memory problems, followed by attentive and executive difficulties, causing impairment to daily living and quality of life [2]. 

Since no effective pharmacological therapy is currently existent for the cure or attenuation of AD’s symptoms and their progression [3], other ways to reduce its burden should be developed. The application of standard cognitive or motor treatments has been one alternative way. The efficacy of motor and cognitive training in patients with AD and Mild Cognitive Impairment (MCI) over six months has been compared in a previous study that showed the positive effects of the two treatments, demonstrating their similar effectiveness in mitigating cognitive decline [4].

In the field of neurorehabilitation, the modulation of cortical excitability and the modification of neuroplasticity have become important mechanisms on which healthcare professionals rely to improve clinical and cognitive functions [5]. Non-Invasive Brain Stimulation (NIBS) techniques have shown great potential in this field. The primary goal of applying NIBS in neurorehabilitation is to modulate cortical excitability in a specific area that supports a specific function in order to facilitate (or suppress) the activity of that area and the interconnected areas [5,6]. NIBS aims to improve the connectivity of the brain network, which, in turn, enhances a particular behavioral or cognitive function associated with that area or network. In recent years, a specific NIBS technique, transcranial direct current stimulation (tDCS), has gained significant public interest in this context [5,7]. tDCS is a method that influences brain activity by using low, continuous electric currents (1–2 mA) applied to the scalp via electrodes for 10–30 min. Depending on the polarity, it can either increase the firing of brain cells and cortical excitability (anodal tDCS) or reduce cortical excitability by making the brain cells less active (cathodal tDCS). The changes in cortical excitability due to tDCS are not limited to the stimulation period and may involve mechanisms similar to those responsible for long-term changes in the brain’s activation [5,7].

Studies have shown that the effects of tDCS last beyond the actual stimulation and can be strengthened through repeated sessions. In the context of neurorehabilitation, tDCS is used to improve cognitive functions by stimulating specific brain networks; however, more studies have tried to apply tDCS in patients with AD [8,9,10]. In the last few years, an increased amount of research has been conducted in this direction on the advantages of the device: its portability, its non-invasive nature, safety, low price, and feasibility to be used in combination with standard therapies [11].

Studies applying tDCS to AD patients differed in their research aims: some focused on memory, mostly on word recognition [12,13,14,15,16] and working memory functions [12]; others on attention [12,13,17] or executive functions [16]; and others on language [14,18,19,20].

Study protocols also varied between each of them for their different methodologies; for example, the targeted areas of stimulation were the left dorsolateral prefrontal cortex [14,17,19,21,22,23,24], the temporoparietal areas [12,15], the left temporal cortex and the right frontal lobe [16], the left angular and supramarginal gyri [18], the left frontotemporal cortex [20], and the bilateral temporal lobes [13,25].

Sensitivity to using tDCS as a rehabilitative tool has increased, but only a few studies, whose sample consisted of patients with dementia, have associated tDCS with a possible treatment [14,18,22,26,27]. Using cognitive training during tDCS can greatly enhance the learning process [28]. This combination is believed to boost brain activity and strengthen specific brain regions involved in cognitive tasks, enhancing long-term brain plasticity by affecting neural networks, with the positive effects extending to similar activities [28,29]. This combined approach might also help the benefits of learning last even after the treatment is over. Previous research has shown that pairing tDCS with cognitive training, specifically targeting the DLPFC, leads to improved learning and performance outcomes, particularly in terms of vigilance measures, across various neurodegenerative disorders [28,30,31,32,33]. Moreover, other research [34] indicates that combining tDCS with cognitive and/or motor training may enhance plasticity, similar to long-term potentiation (LTP), in the targeted region more than the activities alone. This research suggests that adding tDCS to cognitive or motor activities, which effects have already been shown beneficially in Fonte and colleagues in 2019 [4], can create an additional plasticity effect, further improving their efficacy.

However, there are too many disparities in the studies’ methodologies, and there are no clear guidelines on which areas to stimulate or whether to use tDCS before, during, or after cognitive or motor treatment [28]. Therefore, it would be useful to add data in this field to enhance the knowledge of tDCS rehabilitation applications.

Thus, the aim of this study is to evaluate the effectiveness of combining tDCS with motor or cognitive activities for cognitive functions in patients with AD. The second aim is to investigate if tDCS has a different impact if it is combined with motor or cognitive activities.

## 2. Materials and Methods

This is a proof-of-concept pilot study and double-blind randomized controlled trial (RCT) comparing the effects of anodal and sham tDCS combined with motor or cognitive activity on cognitive performance in AD patients. The research team was composed of an “evaluator” and “treatment providers”. The neuropsychologist evaluator was responsible for measuring outcomes and was unaware of the group assignments. Additionally, to assess the presence of a placebo effect or the absence of an effect from the anodal stimulation and to reduce the potential for bias, the participants were also kept unaware of the type of stimulation being administered. In contrast, the treatment providers, including neuropsychologists and physiotherapists, were informed about the treatment, but didn’t know the type of tDCS stimulation used (anodal or sham). Patients and their caregivers were informed about the experimental nature of the study and gave their written informed consent, which was carried out in accordance with the Helsinki Declaration and approved by the local Ethics Committee of the Department of Neuroscience, Biomedicine and Movement Sciences, University of Verona (Protocol 140158) on 5 June 2017. Trial registration ID: NCT06619795 (https://clinicaltrials.gov/) [35].

### 2.1. Participants

Patients with AD were recruited from the Mons. A. Mazzali Foundation (Mantua) between June 2018 and December 2019. The inclusion criteria were as follows: Mini Mental State Examination (MMSE) > 15; good level of compliance; acetylcholinesterase inhibitor treatment (e.g., donepezil, rivastigmine); no modifications of medication during the last four months. The exclusion criteria were as follows: behavioral disorders (e.g., aggressiveness); alcohol abuse; orthopedic pathology with risk of falls to the ground; respiratory pathology; severe uncorrected auditory or visual deficits; history of epileptic fits; anti-epileptic medication; metallic body implants; pacemaker; psychiatric, neurologic, systemic, or metabolic disorders.

According to a simple software-generated randomization scheme (www.randomization.org), after baseline evaluation, patients were allocated to one of the four groups. Unbalanced randomization was applied. The randomization list was not disclosed to the assessors and to the treatment providers, but was known only to the principal investigator (CF).

The procedure for recruiting participants is summarized in Figure 1.

### 2.2. Treatment Procedures

Real tDCS groups received anodal stimulation plus motor activity (MotA) or cognitive activity (CogA). Sham tDCS groups received sham stimulation plus motor activity (MotS) or cognitive activity (CogS).

Each group undertook 45 min of activity, of which the first 15 min involved simultaneous tDCS stimulation. Treatments (45 min) were delivered for two weeks, five times a week.

During the study, participants did not undertake any other motor or cognitive activities.

#### 2.2.1. Motor Activity

We established a standardized sequence of motor exercises. A physiotherapist conducted individual motor activity, which included moderate intensity endurance and resistance training. Sessions started with 5 min of warm-up and ended with 5 min of cool-down, which included active joint mobilization. Subsequently, patients were subjected to endurance exercises, divided into cycling on a cycle ergometer, walking on a platform, and arm cranking on a specific ergometer in a random order [4].

#### 2.2.2. Cognitive Activity

A neuropsychologist conducted individual cognitive activity, which was based upon stimulation of residual cognitive skills and, in particular, of memory. The stimulation was adjusted according to the severity of the cognitive decline observed. Each session began with an orientation exercise. After that, oral and paper–pencil exercises of specific cognitive functions were proposed. These exercises aimed to start the natural process of reminiscence, but they also focused on the present situation, having an impact on social interaction and mood. Multisensory stimulation was introduced.

The structure of the neuropsychological treatment was modeled after a previous study conducted by Spector et al. in 2003, which involved 201 patients with dementia. This approach demonstrated positive outcomes in various cognitive domains and showed improvements in indicators such as quality of life, while maintaining a good level of feasibility and adaptability [4,36].

#### 2.2.3. tDCS Procedure

The tDCS stimulation was performed using a BrainSTIM stimulator (EMS). A pair of saline-soaked electrodes were placed and secured on patients. The anode (25 cm^2^, 5 × 5) was positioned over the left dorsolateral prefrontal cortex (DLPFC) following the 10–20 system (F3–F7 position), and the cathode (35 cm^2^, 7 × 5) was placed above the shoulder on the other side. We adapted our stimulation procedure to align with the prevalent literature, as the DLPFC montage proved to be the most common [28,37]. The intensity of stimulation was set at 2 mA and applied for 15 min at the beginning of the cognitive or motor activity (45 min). The electrode placement and part of the stimulation procedure were adapted from the protocol by Cotelli and colleagues in 2014 [22]. To ensure safety, the current density of the active electrode was kept below the limits established by Poreisz and collaborators in 2007 and Nitsche and collaborators in 2008. In the sham condition, the current was turned off 10 s after the stimulation began and turned back on during the last 10 s, so that the participant could not discern whether the stimulation was real or sham [38,39].

### 2.3. Evaluation Procedures

Primary and secondary outcomes were measured by the same blinded examiner. Patients were evaluated before treatment (T0), immediately after treatment (T1), and after one week in a follow-up (T2). The cognitive assessment was carried out in one day for 1 h. The tests used in the assessment are described below.

#### 2.3.1. Primary Outcome

The primary outcome was the Mini Mental State Examination (MMSE) used to assess the global cognitive impairment [40]. Although the MMSE is a screening test, it has been used as a primary outcome measure in studies on AD [41] (Range: 0–30; higher score = best performance). Across participants whose cognitive ability ranged from normal to moderate–severe AD dementia (n = 15,891), on average, a 1–3 point decrease in MMSE was indicative of a meaningful decline [41].

#### 2.3.2. Secondary Outcome

Picture Recognition (PR) is a subtest of the Rivermead Behavioral Memory Test-3, an ecological memory battery resembling everyday tasks, with the aim to measure daily memory function. The examinee is shown a set of pictures and then is asked to recognize them from a further set of pictures at a later time in the testing session. It has two parallel versions for monitoring changes over time [42] (Range: 0–15; higher score = best performance).

Digit Span Test—Forward (DSF), used to measure span of immediate verbal recall. The examiner presents digits verbally at a rate of one per second. Examiner requires the participant to repeat the digits in the same order. The t of digits increases by one until the participant consecutively fails two trials of the same digit span length (higher score = best performance) [43].

Digit Span Test—Backward (DSB), used to measure working memory. The examiner presents digits verbally and repeats the digits in reverse order. The number of digits increases by one until the participant consecutively fails two trials of the same digit span length [43] (higher score = best performance).

Phonemic Fluency Test (PFT), used to measure processing speed, language production, and executive functions. Participants are given one minute to produce as many unique words as possible starting with a given letter. The participant’s score in each task is given by the number of correct words [44] (higher score = best performance).

Visual Search Test (VST), to assess visual-selective attention. Three matrices are shown to the subject and the patients has to cross the target stimuli between distractors in 45 min [45] (Range: 0–60; higher score = best performance).

Sustained Attention to Response Test (SART), used to evaluate sustained attention and control inhibition. In the test, participants view a computer monitor on which a random series of single digits are presented at the regular rate of 1 per 1.15 s. The task is to press a single response key following each presentation with the exception of a nominated no-go digit, to which no response should be made. In the standard version of the test, 225 digits are presented in a continuous sequence over 4.3 min. The outcomes are false alarm (FA), omission (OM), and reaction time (RT) (higher score = worst performance) [46].

Neuropsychiatric Inventory (NPI) to evaluate the presence, frequency, and severity of behavioral disorders [47] (Range 0–144; higher score = worst performance). The NPI-Q MCID ranges were 2.77 to 3.18 for severity and 3.10 to 3.95 for distress. Residents in the highest NPI-Q tertile at baseline had higher MCID severity (3.62) and distress (5.08) scores than those in the lowest tertile severity (2.40), distress (3.10) [48].

### 2.4. Statistical Analysis

We assessed data from all randomized patients.

The Kruskall–Wallis test was used to measure the homogeneity at baseline for all outcome measures in the four groups.

The Mann–Whitney U test was used to examine the effects of tDCS, comparing MotA + CogA vs. MotS + CogS between T1 and T0 (T1–T0) and between T2 and T0 (T2–T0).

Further analysis was conducted to investigate if there was difference between MotA and CogA groups in performance between T1 and T0 (T1–T0) and between T2 and T0 (T2–T0).

The alpha level for significance was set at *p* < 0.05. Statistical analysis was carried out using the Statistical Package for Social Science (SPSS) for Macintosh, version 26.0 (IBM SPSS Inc., Armonk, NY, USA).

To determine the sample size, based on the study of Khedr and colleagues, an effect size of 1.15 was calculated using the G*Power 3.1.9.4 software. With an effect size of 1.15, an alpha of 0.01, and a power of 0.90, a total of 48 patients were needed to detect a significant treatment effect [25].

## 3. Results

Twenty-three subjects (10 men and 13 women; mean age 83 years) presenting with AD (mean time since onset 3.4 years) were recruited from a total of 314 inpatients attending the Mons. A. Mazzali Foundation (Mantua) (Figure 1). They were randomized into four groups: two real tDCS groups (MotA group n = 6, CogA group n = 7) and two sham tDCS groups (MotS group n = 6, CogS group n = 4). Following the first assessment, there were three dropouts (MotA = 1, MotS = 2), which were not included in the data analysis and were, therefore, excluded from the research.

Patients’ demographic and clinical characteristics are reported in Table 1. Age, education, and MMSE were not statistically different between the four groups. Primary and secondary outcome measures did not significantly differ between the four groups at the baseline (T0).

### 3.1. Primary Outcome’s Results

The Mann–Whitney between-groups comparisons MotA + CogA vs. MotS + CogS showed significant change in MMSE at T1–T0 (*p* = 0.042; z = −2.029). No significant difference was found at T2–T0 (*p* = 0.697; z = −0.389) (see Table 2). For a graphical representation of the results, see Appendix A in the Appendix A.

No significant difference in MMSE was found between MotA vs. CogA groups at T1–T0 (*p* = 0.098; z = −1.653) and at T2–T0 (*p* = 0.409; z = −0.825) (see Table 3).

### 3.2. Secondary Outcome’s Results

Significant differences in the VST (*p* = 0.012, z = −2.515) (see Appendix A in the Appendix A) and SART-FA (*p* = 0.012; z = −2.504) (see Appendix A in the Appendix A) were found between MotA + CogA vs. MotS + CogS at T1–T0 but not at T2–T0 (see Table 2).

Focusing on the effects of the anodal stimulation, no significant difference was found between the MotA vs. CogA at T1–T0. The same analysis showed a significant difference at T2–T0 in PR (*p* = 0.027; z = −2.212) and SART-RT (*p* = 0.047; z = −1.989) (see Table 3).

## 4. Discussion

At present, there is an ongoing debate in the scientific literature regarding whether tDCS exhibits clear therapeutic benefits for patients with neurodegenerative disorders. Furthermore, the current scientific literature still has a limited number of studies examining the efficacy of this combination, with the goal of providing further evidence on the utility of tDCS when paired with targeted rehabilitation interventions [28].

On these bases, our pilot study, building on the work of Cotelli and collaborators in 2014 [22] and Fonte and collaborators in 2019 [4], aimed to explore the effects of simultaneously combining tDCS with cognitive or motor activities on cognition in patients with Alzheimer’s disease. In a study published by our team [4], indeed, it was shown that a motor intervention was comparable effective as the established gold standard treatment for Alzheimer’s disease, which is cognitive stimulation. In the present study, we introduce tDCS stimulation, which occurred during the initial 15 min of either the cognitive or motor stimulation.

Changes in performance of various neuropsychological tests between anodal tDCS and sham tDCS were analyzed.

In the first instance, we investigated the effects of a real stimulation, combining the groups that received real stimulation (MotA + CogA) and comparing them to those who received a placebo stimulation (MotS + CogS).

The results show a significant improvement from baseline to post-treatment in global cognitive status (MMSE) in the real stimulation groups (MotA + CogA). This observed increase could be significant in the context of AD progression and in relation to the “Minimal Clinically Important Difference” (MCID). The MCID represents the minimum change in a score that can be considered a clinically relevant improvement or deterioration in the patient’s quality of life. A study by Andrews and colleagues in 2019 [41] analyzed the MCID for the MMSE, Clinical Dementia Rating Scale, and Functional Activities Questionnaire, showing that a decrease of 1–3 points in the MMSE is indicative of a significant decline in cognitive functions, a decline that tends to be more pronounced as the severity of the disease increases. Consistent with these findings, our study observed a three-point increase in the MMSE between T0 and T1, despite the high variability in results, likely due to the small sample size [42].

The results obtained from this study also show that anodal stimulation, combined with cognitive or motor activities, improves performance in Visual Selective Attention (VST) and inhibitory control (fewer false alarms in SART-FA) at T1–T0 but not at T2–T0. Therefore, the groups receiving anodal stimulation seem to have better performance than the groups receiving sham stimulation [21]. Research on the effectiveness of simultaneous tDCS combined with other treatments is limited, but existing evidence suggests that this approach could be particularly effective in improving cognitive functions and slowing disease progression by increasing neuronal activation and synaptic plasticity [49].

A review conducted by Pilloni and colleagues in 2022 highlighted that tDCS can lead to lasting changes in brain excitability and neuroplasticity, especially when applied in multiple sessions. Studies show that protocols involving repeated sessions produce more significant behavioral changes compared to single or limited-session protocols, with treatment durations in the literature ranging from two days to eight months [28]. Despite the positive effects of tDCS that can emerge after just two or three sessions, uncertainties remain regarding the long-term sustainability of these benefits [28,50]. In our pilot study, a 10-session protocol suggested a positive effect on cognition immediately after treatment; however, for long-term maintenance (follow-up), a longer treatment may be necessary.

Our results reinforce the data present in the literature, whereby AtDCS on the DLPFC area improves the cognitive status [21]. In the literature, several studies indicate the effectiveness of DLPFC stimulation on cognitive functions [28,30,31,32], particularly in recognition memory and overall cognitive function in patients with Alzheimer’s disease [17,21]. This is indeed the most common tDCS setup found in the literature [51]. Activation of the left DLPFC, involved in various cognitive functions such as executive control and memory, plays a key role in self-initiating the use of mnemonic strategies and consolidating information for the formation of long-term memory traces [24]. In order to contribute further results to the literature, we have adapted our stimulation procedure to align with the prevalent literature [28,50].

The second aim is to investigate if anodal tDCS has a different impact if it is combined with motor (MotA) or cognitive activities (CogA). The analysis, globally, did not reveal the superiority of cognitive or motor intervention. Indeed, no changes were found between groups in the baseline post-treatment comparison (T0–T1). This result is in line with previous studies that indicate a similar effect of motor and cognitive activities on cognition [4,52,53]

On the other hand, from the analysis, it emerged that the two groups at follow-up (T2) performed differently in two tests.

Specifically, the MotA group at follow-up performed significantly better in the memory recognition (PR) than the CogA group. This result is partially confirmed by the study of Boggio and collaborators in 2009 [17], which reports that a stimulation of three sessions of AtDCS on the DLPFC (intensity of 2 mA for 30 min) improves performance in tasks of recognition memory [17]. Similarly, the results of our study show that AtDCS combined with motor activity induces an improvement in recognition memory but only one week after the end of treatment.

Conversely, the CogA group, compared with MotA, demonstrated a better performance in reaction time to a stimulus (SART-TR). This result is partially in line with the findings of Stonsaovapak and colleagues [54], which indicate an improvement in performance at the end of the treatment in groups combining cognitive stimulation with AtDCS [54].

Our results show an improvement at T2 but not at T1 for the MotA group compared to the CogA group. Some possible explanations for this result could be the small sample size of our study or the short duration of the follow-up.

## 5. Conclusions

In conclusion, the results of this study indicate a superiority of the AtDCS groups over the StDCS groups in cognitive performance, confirming findings in the existing literature [13,17,20,21]. Comparing the real tDCS groups (MotA and CogA), it was observed that after the rehabilitation (T1), the effects of motor and cognitive activities were similar, with no clear advantage of one treatment over the other.

Our results seem to be in line with studies by Gangemi and collaborators in 2021 [20] and Yu and colleagues in 2021 [10], which respectively support the effectiveness of anodal tDCS in slowing down the progression of Alzheimer’s disease in both the short and long term, and as a convincing instrument in cognitive enhancement and maintenance. On the other hand, in a study by Pellicciari and Miniussi in 2018 [7], the effectiveness of tDCS was not recorded in neurodegenerative patients. A possible explanation for these discrepancy in outcomes would be hypothesizing in the different methodologies applied. The tDCS protocols indeed varied between each other in terms of the stimulated area and the duration of stimulation, the electrode shape and amplitude, the position of reference electrode, the on-line or off-line treatment during the stimulation with tDCS, or the use of underpowered sample sizes [7]. Finally, the assessment procedures could lead to very different results [37]. NIBS techniques may enhance clinical recovery by facilitating functional and structural neuronal changes, strengthening synapses, and increasing dendritic connections [51]. tDCS can alter resting membrane potentials, either enhancing or decreasing underlying cortical excitability. Moreover, tDCS could promote rehabilitation by increasing adaptive neuroplasticity and reducing the pathological consequences in different neurological diseases [34].

This study is one of the pioneering efforts to explore the beneficial effects of combining tDCS with a stimulation in Alzheimer’s patients. In the past, different authors have already investigated the effects of tDCS combined with cognitive stimulation [14,22,26,27], but nobody has explored the effects of tDCS with motor stimulation in these patients.

It is important to note that this study has some limitations, most notably the relatively small participant sample. Based on the sample size calculation, we should have recruited 48 patients; however, 23 patients were recruited in our study. Our sample size was smaller than the post hoc calculated sample size because there were difficulties with the compliance of patients with Alzheimer’s disease. Additionally, the medical issues of these patients and the SARS-CoV-2 infection further decreased the number of participants, preventing us from continuing with their recruitment.

Nevertheless, despite these limitations, the results indicate that 10 sessions of combined tDCS treatment alongside either cognitive or motor activities lead to a more pronounced slowing of cognitive decline when compared to a single administration of either treatment alone.

Since tDCS is not a neurorehabilitation strategy, but modulates brain excitability to simplify the recruitment of brain networks that have been weakened by a diminished synaptic activity [7], it should be paired with rehabilitation protocols to facilitate a more ecological improvement.

This study opens the door to a new perspective in the neurorehabilitation field that allows us to experience tDCS as an additional tool that can amplify the effects of a stimulation, with the aim of improving cognition and slowing decline.

## Figures and Tables

**Figure 1 brainsci-14-01099-f001:**
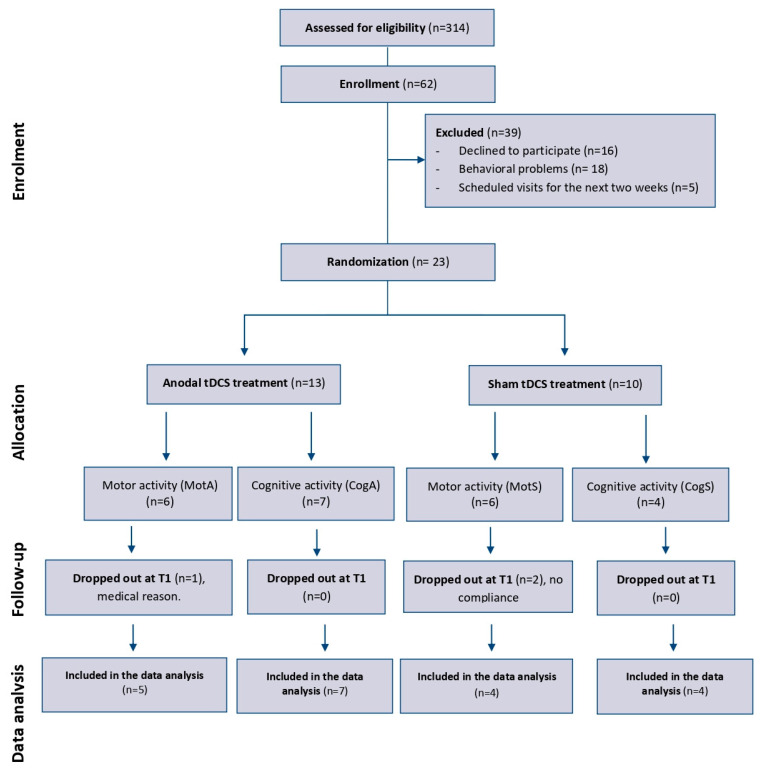
Flow diagram. Representation of the study structure. Abbreviations: MotA: motor activity + anodal stimulation; MotS: motor activity + sham stimulation; CogA: cognitive activity + anodal stimulation; CogS: cognitive activity + sham stimulation.

**Table 1 brainsci-14-01099-t001:** Demographic and clinical data. Data are given as mean ± standard deviation. Abbreviations: MotA: motor activity + anodal stimulation; MotS: motor activity + sham stimulation; CogA: cognitive activity + anodal stimulation; CogS: cognitive activity + sham stimulation; MMSE: Mini Mental State Examination.

	CogA	CogS	MotA	MotS	Baseline Comparison *p* Value
Numbers	7 (3♂/4♀)	4 (2♂/2♀)	6 (2♂/4♀)	6 (2♂/4♀)	
Age (years)	78 ± 11	81 ± 5	85 ± 5	88 ± 3	0.195
Education (years)	8 ± 3	8 ± 3	8 ± 4	8 ± 4	0.892
Time from onset (years)	3 ± 2	3 ± 4	4 ± 4	4 ± 2	0.753
MMSE (0–30)	19.7 ± 4.7	20.7 ± 6.2	19.6 ± 3.6	18 ± 2.5	0.947
Pharmacological treatment					
Cholinesterase inhibitors	4	1	4	3	
Antipsychotics	0	1	2	0	
Antidepressant	4	1	3	1	
Benzodiazepines	2	0	2	2	
Hypertension medications	4	3	3	1	
Proton pump inhibitors	2	2	1	2	
Cholesterol medications	1	1	2	1	
Diuretics	0	1	1	2	
Comorbidity					
Hypertension	4	2	1	1	
Diabetes	1	1	0	0	
Hepatic steatosis	1	0	0	0	
Cholesterol	1	1	1	0	
Cardiovascular diseases	0	0	1	1	
Depression	0	0	1	1	

**Table 2 brainsci-14-01099-t002:** Between-group comparisons of MotA + CogA vs. MotS + CogS. Data are given as mean ± standard deviation. Abbreviations: MotA: motor activity + anodal stimulation; MotS: motor activity + sham stimulation; CogA: cognitive activity + anodal stimulation; CogS: cognitive activity + sham stimulation; MMSE: Mini Mental State Examination; PR: Picture Recognition; DSF: Digit Span—Forward; DSB: Digit Span—Backward; PFT: Phonemic Fluency Test; VST: Visual Search Test; SART-FA, SART-RT, SART-OM: false alarm (FA), reaction time (RT), and omission (OM) of Sustained Attention to Response Test; NPI: Neuropsychiatric Inventory. The “*” symbol indicates results that were statistically significant.

	Real tDCS Groups MotA + CogA		Sham tDCS Groups MotS + CogS		Between-Group Comparisons	
	T1–T0	T2–T0	T1–T0	T2–T0	T1–T0 *p* Value (Z)	T2–T0 *p* Value (Z)
MMSE (0–30)	3 ± 2.17	1.58 ± 2.43	−0.25 ± 3.61	1.12 ± 3.18	0.042 (−2.029) *	0.697 (−0.389)
PR (0–15)	3.25 ± 3.72	1.58 ± 4.12	2.12 ± 2.23	−0.87 ± 2.75	0.558 (−0.585)	0.212 (−1.247)
DSF	−0.16 ± 0.94	−0.33 ± 1.07	−0.5 ± 0.75	−0.37 ± 0.74	0.439 (−0.774)	0.738 (−0.335)
DSB	0.33 ± 0.78	−0.08 ± 0.67	−0.12 ± 0.83	−0.25 ± 1.28	0.242 (−1.169)	0.537 (−0.617)
PFT	7.08 ± 3.34	3.5 ± 11.71	2 ± 7.54	2.62 ± 8.12	0.062 (−1.864)	0.817 (−0.232)
VST (0–60)	5.58 ± 2.87	−0.25 ± 6.55	−0.37 ± 8.81	1.25 ± 5.52	0.012 (−2.515) *	0.642 (−0.456)
SART-FA	−0.7 ± 2.36	0.82 ± 4.64	6 ± 6.05	3.57 ± 7.59	0.012 (−2.504) *	0.441 (−0.771)
SART-RT	−43 ± 89.01	5.27 ± 135.14	10 ± 125.41	57.14 ± 142.39	0.354 (−0.928)	0.683 (−0.408)
SART-OM	−38.9 ± 46.17	−32.09 ± 51.75	−47.14 ± 67.58	−47.28 ± 79.46	0.495 (−0.683)	0.618 (−0.498)
NPI (0–144)	−4.82 ± 6.01	−5.83 ± 6.98	−1 ± 1.07	−7.12 ± 7.59	−112 (−1.591)	0.846 (−0.194)

**Table 3 brainsci-14-01099-t003:** Between-group comparisons of MotA vs. CogA. Data are given as mean ± standard deviation. Abbreviations: MotA: motor activity + anodal stimulation; MotS: motor activity + sham stimulation; CogA: cognitive activity + anodal stimulation; CogS: cognitive activity + sham stimulation; MMSE: Mini Mental State Examination; PR: Picture Recognition; DSF: Digit Span–Forward; DSB: Digit Span—Backward; PFT: Phonemic Fluency Test; VST: Visual Search Test, SART-FA, SART-RT, SART-OM: false alarm (FA), reaction time (RT), and omission (OM) of Sustained Attention to Response Test; NPI: Neuropsychiatric Inventor. The “*” symbol indicates results that were statistically significant.

	MotA		CogA		Between-Group Comparisons	
	T1–T0	T2–T0	T1–T0	T2–T0	T1–T0 *p* Value (Z)	T2–T0 *p* Value (Z)
MMSE (0–30)	4.2 ± 2.58	0.8 ± 2.86	2.14 ± 1.46	2.14 ± 2.11	0.098 (−1.653)	0.409 (−0.825)
PR (0–15)	5.2 ± 4.71	4.8 ± 3.56	1.85 ± 2.26	−0.71 ± 2.81	0.140 (−1.477)	0.027 (−2.212) *
DSF	−0.2 ± 0.44	−0.8 ± 0.83	−0.14 ± 1.21	0 ± 1.15	1.0 (0.000)	0.265 (−1.116)
DSB	0.4 ± 0.89	0.2 ± 0.83	0.28 ± 0.75	−0.28 ± 0.48	0.927 (−0.091)	0.234 (−1.190)
PFT	8.4 ± 2.79	10 ± 14.10	6.14 ± 3.57	−1.14 ± 7.64	0.220 (−1.227)	0.165 (−1.388)
VST (0–60)	4.4 ± 2.70	−0.8 ± 2.58	6.42 ± 2.87	0.14 ± 8.59	0.412 (−0.821)	0.684 (−0.407)
SART-FA	−0.25 ± 3.30	−0.25 ± 4.19	−1 ± 1.78	1.42 ± 5.09	0.828 (−0.217)	0.569 (−0.570)
SART-RT	−15.25 ± 66.44	115 ± 132.63	−61.5 ± 102.89	−57.42 ± 95.02	0.670 (−0.426)	0.047 (−1.989) *
SART-OM	−39.25 ± 19.17	−9.75 ± 61.16	−38.66 ± 60.13	−44.85 ± 45.49	0.522 (−0.640)	0.345 (−0.945)
NPI (0–144)	−6 ± 8.45	−7.2 ± 9.14	−3.83 ± 3.54	−4.85 ± 5.55	1.0 (0.000)	0.684 (−0.407)

## Data Availability

The data presented in this study are available on request from the corresponding author due to privacy reasons.

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
