# Peer review of "Combined Effect of tDCS and Motor or Cognitive Activity in Patients with Alzheimer’s Disease: A Proof-of-Concept Pilot Study"

_brainsci, 2024, doi:10.3390/brainsci14111099_

Round 1
Reviewer 1 Report
Comments and Suggestions for Authors
Dear Authors,
Thank you, authors, for submitting your manuscript for review. I appreciate the effort and time invested in this research. Please see my comments and suggestions to help improve the clarity and robustness of your study:
Introduction:
- The article mentions 50 million individuals affected globally in 2019, but later states 55 million for the same year. This inconsistency needs to be resolved for clarity and accuracy. Additionally, please make uniform the use of anodic/anodal term.
Materials and Methods:
- The choice of a single-blind RCT is appropriate but should be justified in more detail. Please explain why a single-blind design was chosen and discuss its advantages and limitations in the context of this study.
- The randomization process is briefly mentioned. Please provide more details on the software used, any stratification factors, and how allocation concealment was maintained.
- The treatment procedures for motor and cognitive activities are described but lack details on standardization. Please explain how consistency was guaranteed across different sessions and participants.
- Please provide a rationale for using the left DLPFC and shoulder montage.
- The blinding procedure mentions only the examiner. Please describe whether participants and other personnel were blinded and explain how blinding was maintained throughout the study.
- Please provide a comprehensive description of the statistical tests, their assumptions, and any post-hoc analyses performed.
- The sample size appears insufficient. Please provide a power analysis to justify the sample size or discuss any feasibility constraints that led to this sample size.
Results:
- The MMSE showed a significant change at T1-T0 (p=0.042) between MotA+CogA and MotS+CogS. Please discuss the effect size, clinical relevance, and any implications of this change in MMSE scores for AD progression.
- Significant differences in secondary outcomes like VST and SART-FA at T1-T0 were observed, but not consistently across all time points. Provide a detailed discussion on these findings and how they relate to the primary outcome.
- Given the small sample size, variability in the outcome measures could significantly impact results. Please discuss any steps taken to control or account for this variability in the analysis.
- The lack of significant differences at T2-T0 for most outcomes suggests that the effects of the intervention may not be sustained. Please provide a detailed discussion on potential reasons for this and how it impacts the overall interpretation of the study's findings.
I suggested additional graphs to improve the results section:
- Primary Outcome (MMSE) Change Over Time Line Graph:
- Please illustrate changes in MMSE scores from baseline (T0) to post-treatment (T1) and follow-up (T2).
- Plot mean MMSE scores with error bars (standard deviation or standard error) for each group over the three time points. Use different lines for each group (MotA, MotS, CogA, CogS).
- Secondary Outcomes Comparison Bar Graph:
- Compare the changes in secondary outcome measures (e.g., PR, DSF, DSB, PFT, VST, SART, NPI) between groups.
- Use grouped bar charts to show mean changes in each secondary outcome measure from T0 to T1 and T2 for each group.
Discussion:
- Discussion mentions differing outcomes in tDCS studies, suggesting methodological variances as a cause. Please expand on how your methodology aligns or diverges from these studies to provide a clearer context.
- While the discussion highlights the effectiveness of DLPFC stimulation, it would benefit from a more detailed comparison with existing research. Please discuss similarities and differences in outcomes and methodologies.
- The discussion notes improvements at T1 but not at T2. This suggests short-term benefits that may not persist. Please discuss potential reasons for this pattern and implications for the durability of tDCS effects.
- Please provide a deeper analysis of why certain measures showed improvement while others did not, and what this means for the overall efficacy of tDCS.
N/A
Author Response
Dear Reviewer, thank you for your suggestions. We have uploaded the Word file where we have commented on and highlighted the corrections within our paper.

Reviewer 2 Report
Comments and Suggestions for Authors
This reviewer thanks the editorial board for the opportunity to review this manuscript and learn about the interesting research activity of its authors. This is a pilot study with a small sample, divided into four intervention groups, with the main objective of to investigate the effects of combined motor or cognitive activity with tDCS on cognitive functions in Alzheimer disease patients. This reviewer raises the following questions that should be clarified:
Major issues:
The objectives of the study should be more rigorous throughout the manuscript, which is why sometimes this reviewer doubts its initial approach. The authors explain in the discussion that they first wanted to investigate the effects of real tDCS vs placebo. To do this, in order to show some positive result, they compare MotA+CogA vs MotS+CogS, adding the results of 2 groups and comparing with another 2. Thus, methodologically it does not seem to make much sense, since in this case the study should have had only two intervention groups.
Despite this, the main objective of the study was “to investigate the effects of combined motor or cognitive activity with tDCS on cognitive functions in Alzheimer disease patients”, which suggests that the authors reformulated the purpose of the study once they saw the results. results obtained.
This problem should be reasonably explained or eliminated from the statistics and text of the manuscript. Furthermore, the study lacks a previous record that could clarify this doubt of the reviewer.
Minor issues:
INTRODUCTION
- The data expressed in lines 33-37 do not make sense among themselves. Please provide updated information.
- Specify in L.82 that these studies deal with dementia patients.
- Adapt the hypothesis to the proposed objective.
- The gap exposed by the authors needs a greater connection with the study developed.
METHODS
- Unify the criteria of whether tDCS and motor/cognitive activity were “combined” or “associated”.
- Which professional was in charge of examining the patient selection criteria?
- Please adapt the Flow diagram to CONSORT guidelines.
- Are the motor/cognitive activities programs based on any previous protocol? Reference.
- Is the cathode placement reasonably placed according to any previous protocol? Reference. If not, reason.
- The authors must explain the procedure they carried out to apply the tDCS sham, reasoning that this procedure can achieve good masking.
- Review the psychometric properties of each of the evaluation questionnaires, as well as the MCID values.
RESULTS
- What was the reason why the groups were not homogeneous in number of subjects?
DISCUSSION
- Reference correctly using the same method for all studies (e.g. Gangemi or Yu).
Comments on the Quality of English Language
The authors must carefully reread the manuscript and correct any typographical errors.
Author Response

(The authors gave the same response as above.)

Round 2
Reviewer 2 Report
Comments and Suggestions for Authors
The authors have carefully reviewed the manuscript and have modified the suggestions presented. They have also justified the results and conclusions they have reached in the study.